# The Association of CD8+ Cytotoxic T Cells and Granzyme B+ Lymphocytes with Immunosuppressive Factors, Tumor Stage and Prognosis in Cutaneous Melanoma

**DOI:** 10.3390/biomedicines10123209

**Published:** 2022-12-10

**Authors:** Satu Salmi, Kaisla Hälinen, Anton Lin, Sanna Suikkanen, Otto Jokelainen, Eija Rahunen, Hanna Siiskonen, Sanna Pasonen-Seppänen

**Affiliations:** 1Institute of Biomedicine, University of Eastern Finland, 70211 Kuopio, Finland; 2Department of Clinical Pathology, Kuopio University Hospital, 70029 Kuopio, Finland; 3Institute of Clinical Medicine/Clinical Pathology, University of Eastern Finland, 70029 Kuopio, Finland

**Keywords:** melanoma, tumor microenvironment, lymphocytes, immunosuppression

## Abstract

The immunosuppressive tumor microenvironment (TME) consists of suppressive cells producing a variety of immunomodulatory proteins, such as programmed death ligand 1 (PD-L1) and indoleamine-2,3-dioxygenase (IDO). Although granzyme B (GrB) is known to convey the cytolytic activities of CD8+ cytotoxic lymphocytes, it is also expressed by other cells, such as regulatory T and B cells, for immunosuppressive purposes. The role of GrB+ lymphocytes in melanoma has not been examined extensively. In this study, benign, premalignant, and malignant melanocytic tumors were stained immunohistochemically for CD8 and GrB. PD-L1 was also stained from malignant samples that had accompanying clinicopathological data. The association of CD8+ and GrB+ lymphocytes with PD-L1 expression, tumor stage, prognosis, and previously analyzed immunosuppressive factors were evaluated. Our aim was to obtain a more comprehensive perception of the immunosuppressive TME in melanoma. The results show that both CD8+ and GrB+ lymphocytes were more abundant in pT4 compared to pT1 melanomas, and in lymph node metastases compared to primary melanomas. Surprisingly, a low GrB/CD8 ratio was associated with better recurrence-free survival in primary melanomas, which indicates that GrB+ lymphocytes might represent activated immunosuppressive lymphocytes rather than cytotoxic T cells. In the present study, CD8+ lymphocytes associated positively with both tumor and stromal immune cell PD-L1 and IDO expression. In addition, PD-L1+ tumor and stromal immune cells associated positively with IDO+ stromal immune and melanoma cells. The data suggest that IDO and PD-L1 seem to be key immunosuppressive factors in CD8+ lymphocyte-predominant tumors in CM.

## 1. Introduction

During the last few years, the prognosis of advanced cutaneous melanoma (CM) has considerably improved due to the development of novel immune checkpoint inhibitor (ICI) therapies, which act to enhance the anti-tumor immune responses of CD8+ lymphocytes [1]. As microenvironment-related factors may interfere and thus promote melanoma cells to resist and escape immunotherapy, it is essential to better understand how an immunosuppressive tumor microenvironment (TME) is formed and maintained in CM [2].

Immunosuppressive features enable melanoma cells to evade the host immune system by suppressing their attack by CD8+ cytotoxic T lymphocytes (CTLs) [3]. The immunosuppressive TME in CM is composed of different immune cells, such as FoxP3+ Regulatory T cells (Tregs) and tumor-associated macrophages (TAMs), by the expression of immunomodulatory proteins, of which programmed death ligand 1 (PD-L1) and indoleamine 2,3-dioxygenase (IDO) are notable examples, as well as different immunomodulatory cytokines and fibroblasts [4].

Despite numerous tumor-infiltrating lymphocytes (TILs), melanoma cells often escape surveillance by the immune system [5]. TILs are known to be a positive prognostic factor in vertical growth phase melanomas [6]. However, the prognostic value of TILs in patients treated with ICIs is unclear [7]. CD8+ CTLs are also associated with a favorable prognosis in several studies [8], both in primary [9] and advanced melanomas [10,11], and may also predict response to ICIs [12]. However, in some studies, CTLs have not been shown to be associated with CM prognosis [13].

Granzyme B (GrB) is a serine protease excreted through exocytosis by CTLs in order to kill tumor cells [14]. Moreover, natural killer cells express GrB [15]. Thus, GrB has been used as a marker for anti-tumor cytolytic activity in, for example, colorectal cancer, where it associates positively with increased survival times [16,17]. However, GrB is also expressed by immunosuppressive cells, such as FoxP3+ Tregs and B regulatory cells (Bregs), to mediate their immunosuppressive effects [18]. In CM, GrB expression has not been studied extensively. One study has found that the absence of GrB+ TILs in primary melanomas was associated with sentinel lymph node metastasis [19].

PD-1 (programmed death receptor 1) is a co-inhibitory cell surface receptor expressed by antigen-stimulated T cells. PD-1 and its ligand, PD-L1, are targets of ICIs for inhibition. Activation of the PD-1/PD-L1 axis leads to the phenomenon of antigen-specific T cell exhaustion. Tumor and stromal cells can also express PD-L1 and thus dampen the anti-tumor immune reaction [20]. The prognostic significance of PD-L1 in CM remains unclear. In some studies, PD-L1+ tumor or stromal cells associate with poor outcome [21,22], whereas in other reports, they have failed to found any association between PD-L1 expression and cancer prognosis [11,23]. The contradictory findings may be, at least partly, explained by the heterogeneity of study material, and by the different antibodies and analyzing methods used.

The purpose of this study was to investigate the associations of CD8+ and GrB+ lymphocytes as well as tumor and stromal PD-L1 expression with tumor stage, prognosis and clinicopathological parameters in CM. Moreover, their association with other immunosuppressive factors (TAMs, FoxP3+ Tregs, IDO) was assessed from the results that were based on the same study material, which was published separately [24,25]. This study focused on CD8+ and GrB+ lymphocytes, because CD8 is a marker for all cytotoxic T cells and this marker does not correlate with lymphocyte activity. In contrast, GrB is produced by activated lymphocytes, which can represent immunosuppressive Tregs or regulatory B cells (Bregs), or cytotoxic CD8+ T cells and natural killer cells [18].

## 2. Materials and Methods

### 2.1. Histological Specimens

This study consists of benign, pre-malignant and malignant melanocytic lesions collected at the Kuopio University Hospital (Finland) during the years 1980–2010. The malignant melanocytic lesions consisted of in situ melanomas restricted to the epidermis, superficial melanomas with less than 1 mm invasion and deep melanomas with more than 4 mm invasion. From a total of 252 samples, 250 were stained for CD8 and 249 for GrB. PD-L1 was stained from 128 malignant tumors for which clinicopathological patient data were available. Samples with large necrotic areas, abundant pigmentation or insufficient tumor area were omitted, yielding 203 CD8, 172 GrB and 68 PD-L1 samples for analysis (Table 1). Analyzed CD8 and GrB stained samples including patient data are shown in Table 2. This study was approved by the research ethics committee of the Northern Savo Hospital District and by the Finnish National Supervisory Authority for Welfare and Health (VALVIRA, 6187/05.01.00.06/2010).

### 2.2. Immunohistochemistry

Tissue samples, 5 µm thick, were formalin fixed and paraffin embedded. For CD8- and GrB-specific immunohistochemical staining of samples, after deparaffinization, antigen retrieval was performed in 10 mM citrate buffer (pH 6.0) in a pressure cooker for 15 min, and the samples were washed with 0.1 M phosphate buffer (PB; pH 7.0). Next, the endogenous peroxidase activity was blocked with 1% H_2_O_2_ for 5 min. Finally, to block the non-specific antibody binding, the sections were washed and incubated with 1% milk powder dissolved in PB for 30 min at 37 °C.

Thereafter, the sections were incubated with rabbit monoclonal antibody for CD8 (1:200, Thermo Fisher Scientific, Rockford, IL, USA) and rabbit polyclonal antibody for granzyme B (1:100, Atlas Antibodies, Bromma, Sweden) at 4 °C overnight. The next day, the sections were rinsed with PB and incubated with biotinylated anti-rabbit antibody (1:200, Vector Laboratories) diluted in 1% milk powder in PB. The residual specific antibody was visualized with the Vectastain Elite ABC kit using DAB as the chromogenic substrate to show positive staining in brown, and Mayer’s hematoxylin was used for counterstain the nuclei in blue. Thereafter, the sections were washed, dehydrated, and mounted in DePex.

For anti PD-L1 stainings, before deparaffinization, the sections were cooked at 58 °C for 30 min. After that, the sections were treated similarly to the CD8 and GrB stained material. Rabbit monoclonal anti-PD-L1 antibody (1:80 dilution, Cell Signalling, Danvers, MA, USA) was used as the primary antibody and the same biotinylated anti-rabbit antibody was used as the secondary antibody. For negative control samples, the primary antibodies were omitted from the procedure. In addition, tonsil-derived sample sections were used as a positive control for CD8 and GrB staining and, for PD-L1 immunohistochemical staining, sample sections derived from the placenta were used as the positive control.

Stainings for CD68 and CD163 have been described in [24]. Stainings for IDO and FoxP3 have been described in [25].

### 2.3. GrB + FoxP3 Immunofluorescence Double Stainings

Antigen retrieval was performed on deparaffinized sections in 10 mM citrate buffer (pH 6.0) in a pressure cooker for 15 min, followed by a wash in 0.1 M phosphate buffer (PB; pH 7.0). Subsequently, to quench any autofluorescence, the sections were treated with 50 mM glycine for 40 min at room temperature. Thereafter, non-specific staining was blocked with 1% bovine serum albumin for 30 min, followed by an overnight incubation at 4 °C with the primary antibodies against FoxP3 (Abcam, Cambridge, UK) and GrB (both antibodies at a 1:90 dilutions). Next, the sections were washed and incubated for 1 h with the secondary antibodies (1:300, Alexa Fluor 594 anti-rabbit IgG, Cell Signaling and 1:300 Alexa Fluor 488 anti-mouse IgG, Cell Signaling). Nuclei were counterstained with DAPI (1 μg/mL, Sigma-Aldrich, St. Louis, MO, USA). Finally, the sections were mounted in Vectashield (Vector H-1000, Vector). The samples were imaged with a Zeiss Axio Observer inverted microscope (×20 or ×40 NA 1.3 oil objectives) equipped with a Zeiss LSM 700 confocal module (Carl Zeiss Microimaging GmbH, Jena, Germany).

### 2.4. Analysis of CD8 and GrB Stained Samples

CD8 and GrB stainings were analyzed using the hotspot method described earlier [24,25]. The slides were scanned with a whole-slide digital scanner (Hamamatsu NanoZoomer S360). The areas with the highest densities of positive cells were chosen from the scanned slides by the Visiopharm-software and the pictures of these hotspots were captured within the software. If the accuracy of the digitalized slide was not high enough, the sample was imaged with a Zeiss AxioCam ERc 5S microscope-mounted camera (Carl Zeiss, Germany) with identical picture sizes. Three or five hotspots were chosen depending on the lesion size.

An automated computed vision (CV) software reported earlier was used to analyze CD8+ lymphocytes from the images [25]. The software was added with the option to manually correct the selections of the CV. First, all cell selections performed by the CV were checked from the images and corrected semiautomatically from the significant outliers (327 pictures, 36% of all pictures). Validation datasets were created separately for the remaining 390 scanned and 186 microscopic images by analyzing semiautomatically 35% of both microscopic and scanned images (S.Sa.). The Pearson correlation coefficients between automated and semiautomated analyses were r = 0.990 and r = 0.970 for microscopic images and scanned images, respectively (*p*-values < 0.001). GrB+ cells were counted manually from the images since CV was not sufficient due to the variable non-specific background staining. GrB+ lymphocytes were distinguished from other GrB-expressing cells by their shape and size.

In total, 25% of both CD8 and GrB stained sections (51 and 43 samples, respectively), containing samples from all histopathological groups, were analyzed independently by another investigator (K.H.) by imaging the hotspots with microscope-mounted camera and counting positive cells from the pictures semiautomatically/manually. The Pearson correlation coefficients between the independently analyzed cell counts were r = 0.964 for CD8 and r = 0.920 for GrB stained sections (*p*-values < 0.001).

### 2.5. Analysis of PD-L1 Stainings

PD-L1 was analyzed from 68 malignant tumors, for which patient data were available. The expression of PD-L1 was evaluated semiquantitatively and separately from stromal immune and tumor cells. A four-level scoring system was used for the assessment. Score 0 was given if <1% of cells were PD-L1 positive. Scores 1 to 3 were given if 1–5%, 6–20% or >20% of cells expressed PD-L1, respectively. PD-L1 expression was evaluated independently by two investigators (S.Sa., H.S.), and samples with differing results were re-evaluated together.

### 2.6. Statistical Analysis

The different statistical analyses were performed using the SPSS Statistics 27 package (IBM Corporation, Armonk, New York, NY, USA). For the comparison between the different histological groups, a non-parametric Kruskal–Wallis test, with pairwise comparisons, was used. A Pearson χ^2^-test was employed to analyze the associations with semiquantitatively assessed immunosuppressive factors and clinicopathological parameters. Univariate and multivariate survival analyses were conducted by using a Kaplan–Meier with log-rank test and a Cox’s regression, respectively. A Mann–Whitney U-test was used to interrogate the association of PD-L1 expression with quantitatively assessed immunosuppressive factors. For the χ^2^-test, Mann–Whitney U-test, and survival evaluations, the CD8+ and GrB+ lymphocyte counts were divided into two groups (low or high) according to the median value. Cell counts less than or equal to the median corresponded to low, and cell counts higher than the median to high cell counts (median 567.60 for CD8 and 79.60 for GrB). PD-L1 expression of tumor and stromal immune cells was also divided into two groups. PD-L1 expression was considered low if ≤5% and high if >5% of cells expressed PD-L1.

## 3. Results

### 3.1. Patient Characteristics

The clinicopathological characteristics of the cohort are shown in Table 3. The mean follow-up was 10.2 ± 9.3 years (median 8.0 years). Patients were diagnosed and followed up in the era that preceded the use of current immunotherapies.

### 3.2. CD8+ and GrB+ Lymphocytes Associate Positively with Breslow’s Depth

To evaluate the density and localization of CD8+ and GrB+ lymphocytes, melanocytic tumors were stained for CD8 and GrB. The staining pattern of CD8 was cytoplasmic and membranous. GrB showed granular, cytoplasmic staining pattern and was also localized to the plasma membrane (Figure 1 and Figure 2). Altogether, deep melanomas showed more CD8+ and GrB+ lymphocytes than superficial melanomas or nevi. Both CD8+ and GrB+ lymphocytes localized mainly in the stromal compartment. In deep melanomas and lymph node metastasis (LNMs) samples, there were also some CD8-positive lymphocytes inside the tumor cell nests (Figure 1E,F).

The number of CD8+ and GrB+ lymphocytes was significantly higher in pT4 compared to pT1 melanomas (*p*-values 0.023 and 0.014, respectively). In addition, both CD8+ and GrB+ lymphocytes were more abundant in LNMs compared to pT4 melanomas (*p*-values 0.011 and 0.013, respectively), as well as in LNMs compared to pT1 melanomas (*p*-values < 0.001) (Figure 3A,B).

In melanocytic tumors, there was a positive correlation between CD8+ and GrB+ lymphocytes (Spearman’s rho r_s_ = 0.829, *p* < 0.001) (Table 4).

### 3.3. The Association of CD8+ and GrB+ Lymphocytes with Clinicopathological Parameters

CD8+ and GrB+ lymphocytes associated positively with disease recurrence (*p*-values < 0.001) as well as poor prognostic factors, such as ulceration (*p*-values < 0.001 and 0.046, respectively) and nodular growth phase (*p*-values < 0.001 and 0.023, respectively) (Table 5).

### 3.4. A Low GrB/CD8 Ratio Is an Independent Positive Prognostic Factor for Non-Immunotherapy-Related RFS in Primary Melanomas

In the group of all malignant lesions, high CD8+ and GrB+ lymphocyte counts associated with poor recurrence-free survival (RFS) in a univariate survival analysis (*p*-values < 0.001 and 0.002, respectively); however, the result was not retained in a multivariate analysis, which took tumor stage as a covariate. In the group of primary melanomas only, high CD8 count associated with poor RFS (*p* = 0.015) and poor disease-specific survival (DSS) (*p* = 0.037) but was not an independent prognostic factor when Breslow’s depth was used as a covariate. In the group of primary melanomas, GrB cell number alone did not associate with survival.

In the group of all malignant cases and primary melanomas only, a GrB/CD8 cell number ratio of <0.1 was associated with a better RFS (*p*-values < 0.001 and 0.019, respectively) and DSS (*p*-values 0.024 and 0.030, respectively), and was an independent positive prognostic factor for RFS in primary melanomas when Breslow’s depth was taken as a covariate in a multivariate analysis of survival (*p* = 0.012, HR: 0.195, 95% CI: 0.54–0.699) (Figure 3C).

### 3.5. The Association of PD-L1 Expression with Tumor Stage and Clinicopathological Parameters

Malignant tumors containing the patient data were immune-histologically interrogated with anti PD-L1 antibodies. Tumors were divided into low (≤5% of cells) and high (>5% of cells) PD-L1-expressing tumors. PD-L1+ tumor and stromal immune cells were significantly more abundant in pT4 compared to pT1 tumors (*p*-values 0.037 and 0.034, respectively) (Table 6).

No statistically significant differences between PD-L1 expression in different tumor stages was observed when PD-L1 was evaluated in a four-level scoring system (Figure 4A,B). In general, the staining pattern of PD-L1 was cytoplasmic and membranous. Positionally, most often, PD-L1+ cells (tumor cells/stromal cells) localized in the tumor–stroma interface (Figure 4C–H).

PD-L1+ stromal immune cells associated positively with ulceration (*p* = 0.018) and both tumor and stromal immune cell PD-L1 expression associated positively with nodular growth phase (*p*-values 0.009 and 0.001, respectively) (Table 6). PD-L1 expression was not found to be associated with survival.

### 3.6. The Association of CD8+ and GrB+ Lymphocytes with Immunosuppressive Factors

The correlations of CD8+ and GrB+ lymphocytes with immunosuppressive factors (PD-L1, IDO, FoxP3 Tregs and TAMs) were evaluated from malignant tumors. CD8+ CTLs associated positively with PD-L1+ tumor and stromal immune cells (*p*-values 0.015 and <0.001, respectively) (Table 6), IDO+ melanoma cells (*p* < 0.001) and tumor nest CD68+ macrophages (*p* = 0.016) (data not shown). There was a moderate positive correlation between CD8+ CTLs and IDO+ stromal immune cells (r_s_ = 0.599, *p* < 0.001), but only weak positive correlations with FoxP3+ Tregs and total macrophage counts (Table 4).

GrB+ lymphocytes associated positively with PD-L1+ tumor and stromal immune cells (*p*-values 0.009 and 0.027, respectively) (Table 6), IDO+ melanoma cells (*p* < 0.001), and tumor nest CD68+ and CD163+ macrophages (*p*-values 0.002 and 0.003, respectively) (data not shown). In addition, there was a moderate positive correlation between GrB+ lymphocytes and IDO+ stromal cells (r_s_ = 0.542, *p* < 0.001) as well as FoxP3 Tregs (r_s_ = 0.479, *p* < 0.001) (Table 4). In immunofluorescence double immunohistochemical staining experiments, whilst GrB+ cells localized to the same region as FoxP3+ Tregs in the tumor–stroma interface, no clear colocalization was observed (Figure 2G).

In conclusion, both CD8+ and GrB+ lymphocytes associated positively with PD-L1+ and IDO+ melanoma and stromal immune cells. Only GrB+ lymphocytes associated with FoxP3+ Tregs and tumor nest CD163+ macrophages.

### 3.7. The Association of Tumor and Stromal Immune Cell PD-L1 Expression with Other Immunosuppressive Factors

PD-L1+ tumor and stromal immune cells associated positively with IDO+ melanoma cells (*p*-values 0.040 and 0.044, respectively) and IDO+ stromal immune cells (*p*-values 0.017 and 0.002, respectively) (data not shown). PD-L1 expression associated neither with FoxP3+ Tregs nor TAMs.

## 4. Discussion

The aim of this study was to investigate the association of CD8+ and GrB+ lymphocytes with tumor stage, survival, and immunosuppressive factors, in order to obtain a more comprehensive perception of the immunosuppressive TME in CM. Currently, there are only a few immunohistochemical studies of GrB+ lymphocytes in CM, and none of these have evaluated the association of GrB+ cells with the tumor stage. Nor have most of the previous melanoma studies of CD8+ CTLs focused on their association with the tumor stage but rather on survival. According to our findings, CD8+ and GrB+ lymphocytes are more abundant in pT4 compared to pT1 melanomas and in LNMs compared to primary melanomas. In the present work, we found that CD8+ CTL count is not an independent prognostic factor for non-immunotherapy-related survival, which is in line with results from Wong and colleagues who found that CD8+ CTLs associate with a favorable prognosis in patients treated with PD-1 inhibition but not in the absence of immunotherapy [26]. Moreover, the present work demonstrates that CD8+ and GrB+ lymphocytes associate positively with PD-L1+ and IDO+ tumor and stromal immune cells in the TME.

Surprisingly, a low GrB/CD8 cell count ratio (<0.1) associated with better non-immunotherapy-related RFS in primary melanomas. In colorectal cancer, GrB+ cells have been shown to associate with better prognosis and this has been used as a cytolytic marker for anti-tumor immunity [16,17]. The present results indicating that a low amount of GrB+ lymphocytes with respect to tumor-infiltrating CTLs associates with better RFS may be an indication of an immunosuppressive role for GrB in CM. Indeed, GrB is also expressed by a variety of immunosuppressive cells, such as Bregs and Tregs [18]. For example, GrB+ B cells have been shown to inhibit the proliferation of T cells by degrading the T cell receptor zeta chain, which is a substrate for GrB, in a GrB-dependent manner [27].

Sabbatino and co-workers analyzed immune cells in thin melanomas and found that GrB+ cells did not colocalize with CD8+ cells in double immunohistological staining experiments [28]. In our study, GrB and FoxP3 did not colocalize either; however, further analysis, in order to specify GrB+ lymphocytes in CM, is needed.

Furthermore, there are no previous immunohistochemical reports of the association of GrB+ lymphocytes with survival in CM. In one study, Wu and colleagues used a single-sample gene set enrichment analysis to assess the role of granzymes in CM and found that GrB associates positively with immunotherapy-related prognosis [29].

In this study, we found that PD-L1 expression was higher in deep compared to thin melanomas, which is in line with previous results [30]. CD8+ CTLs associated positively with PD-L1+ tumor and stromal immune cells, which also corresponds to previous findings [31,32]. Furthermore, CD8+ CTLs associated positively with IDO+ stromal immune and melanoma cells and tumor nest CD68+ macrophages, but only weakly with FoxP3+ Tregs and total macrophage counts. In contrast, in their study, Spranger and co-workers found a strong positive correlation between CD8+ CTLs and FoxP3+ Tregs in melanoma metastases [33]. However, another study found that Tregs in melanoma metastases had a significantly lower immunosuppressive phenotype compared with Tregs in ovarian tumors [34]. Thus, the role of Tregs in CM seems to be conflicting, and presumably their immunosuppressive role is influenced by other components of the TME. Furthermore, according to our results, PD-L1+ stromal immune and melanoma cells associated positively with IDO+ stromal immune and tumor cells, but not with TAMs or FoxP3 Tregs. Similarly, Spranger and colleagues have reported a strong positive correlation between the expression of IDO and PD-L1 in CD8+ lymphocyte-predominant metastases [33]. The present results suggest that PD-L1-and-IDO-mediated immunosuppression might be especially important in CD8+ CTL inactivation, as they clearly accumulate in CD8+ lymphocyte-rich tumors.

We also found that GrB+ lymphocytes associate positively with both IDO+ melanoma cells and tumor nest macrophages and moderately with FoxP3 Tregs and IDO+ stromal immune cells. To our knowledge, similar associations have not been studied before in CM. Our results suggest that GrB+ lymphocytes might represent mainly immunosuppressive lymphocytes and thus the positive association of GrB+ lymphocytes with different immunosuppressive factors may indicate that the activation of immunosuppressive lymphocytes is associated with a concurrent accumulation of other immunosuppressive factors into the tumor. However, further studies are needed to examine the role of GrB+ lymphocytes in CM.

In the present study, both CD8+ and GrB+ lymphocyte counts associated positively with tumor stage. Higher numbers of these cells also correlated with poor clinicopathological factors, such as recurrence and ulceration. It is likely that although the number of CD8+ CTLs is higher in more advanced tumors, their activation status is decreased in deep melanomas and LNMs compared with thin melanomas. Indeed, negative immunoregulation, for example, gene expression signatures associated with exhausted T cells, has been shown to progressively increase from benign nevi, dysplastic nevi through to malignant melanoma [35]. In addition, if the role of GrB+ lymphocytes in CM would be mainly immunosuppressive, as we have hypothesized based on our findings, it would be logical that these activated cells associate positively with tumor stage and poor prognostic factors.

In conclusion, our results suggest that the amounts of tumor-infiltrating CD8+ and GrB+ lymphocytes increase with the tumor malignancy. IDO and PD-L1 seem to be key immunosuppressive factors in CD8+ lymphocyte-predominant tumors. However, GrB+ lymphocytes seem to represent the cytolytic activity of immunosuppressive lymphocytes rather than CTLs, and their amount appears to associate with the accumulation of several other immunosuppressive cells into the tumor.

## Figures and Tables

**Figure 1 biomedicines-10-03209-f001:**
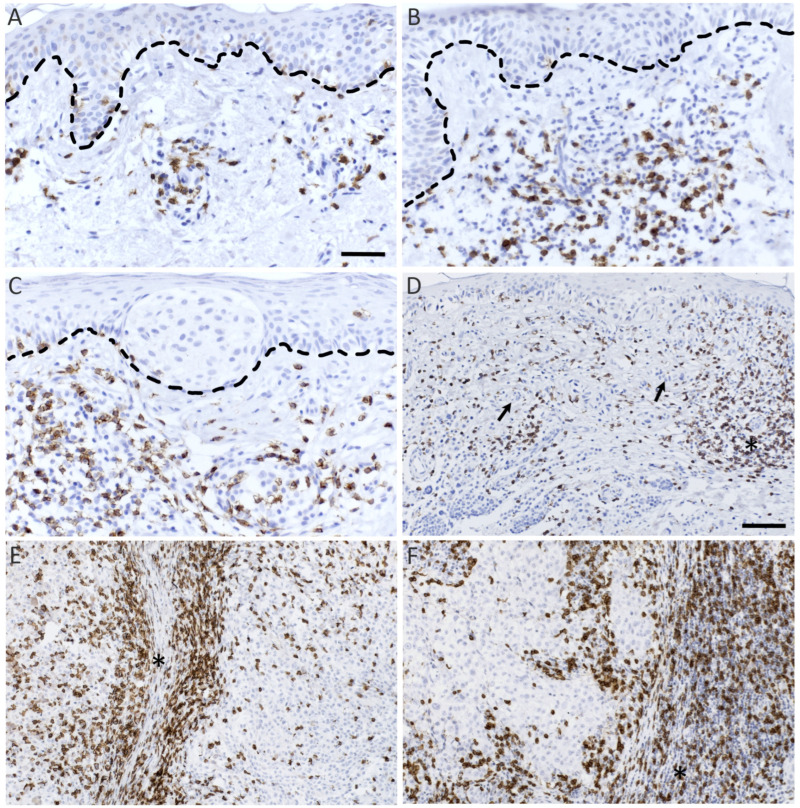
Representative immunohistochemical stainings of CD8+ lymphocytes (positive staining shown in brown and counterstained nuclei in blue). Immunohistochemical stainings of CD8 in benign (**A**) and dysplastic nevi (**B**), in situ melanoma (**C**), superficial (Breslow’s depth < 1 mm, (**D**)) and deep melanomas (Breslow’s depth > 4 mm, (**E**)) and lymph node metastasis (**F**). The dashed line in (**A**,**B**) defines the epidermis in benign and dysplastic nevi, respectively, and the dashed line in (**C**) marks the tumor–stroma borderline in in situ melanoma. The asterisks in (**D**–**F**) indicates the stromal tumor compartment, and black arrows in (**D**) point to tumor cells. Scale bar is 50 μm in (**A**) (and applies across (**A**–**C**), which all have ×200 magnification) and 100 μm in (**D**) (which applies across (**D**–**F**), which have all the same ×100 magnification), respectively.

**Figure 2 biomedicines-10-03209-f002:**
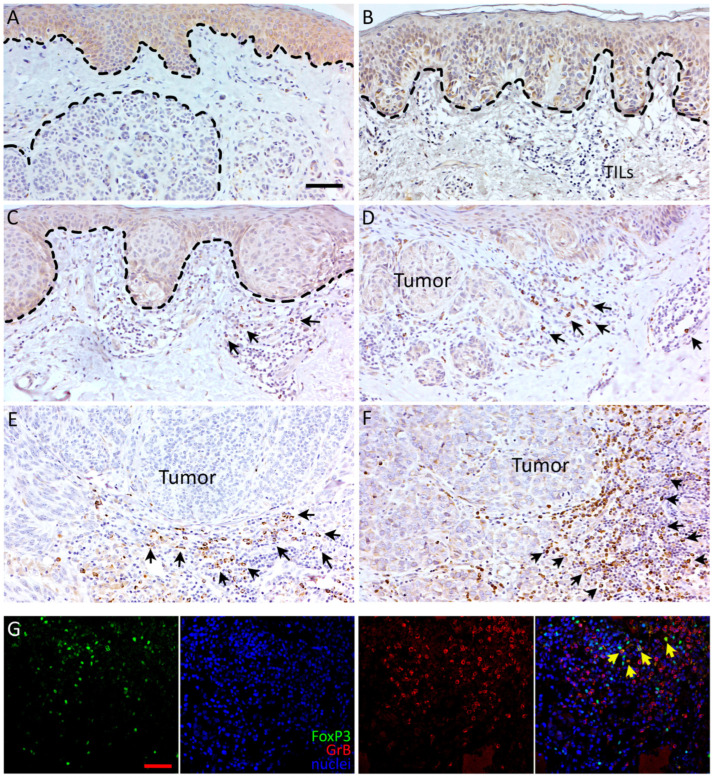
Representative immunohistochemical detection of GrB+ lymphocytes (positive staining shown in brown and counterstained nuclei in blue in (**A**–**F**)). Immunohistochemical stainings of GrB in benign (**A**) and dysplastic nevi (**B**), in situ melanoma (**C**), superficial (Breslow’s depth < 1 mm, (**D**)) and deep (Breslow’s depth > 4 mm, (**E**)) melanomas and lymph node metastasis (**F**). The dashed line in (**A**,**B**) marks the epidermis in benign and dysplastic nevi, respectively, and intradermal nest of nevus cells in (**A**). The dashed line in (**C**) defines the border between tumor and stroma in in situ melanoma. The arrows in (**C**–**F**) point to the stromal GrB+ cells. Scale bar is 50 μm in (**A**) applies for (**A**–**F**) (×200 magnification). Immunofluorescence double staining of GrB and FoxP3 in pT4 melanoma (**G**) indicates that GrB+ cells localize in the same region as FoxP3+ Tregs in the tumor–stroma interface but no clear colocalization is observed (arrows). Scale bar 50 µm (×100 magnification). TILS = tumor-infiltrating lymphocytes.

**Figure 3 biomedicines-10-03209-f003:**
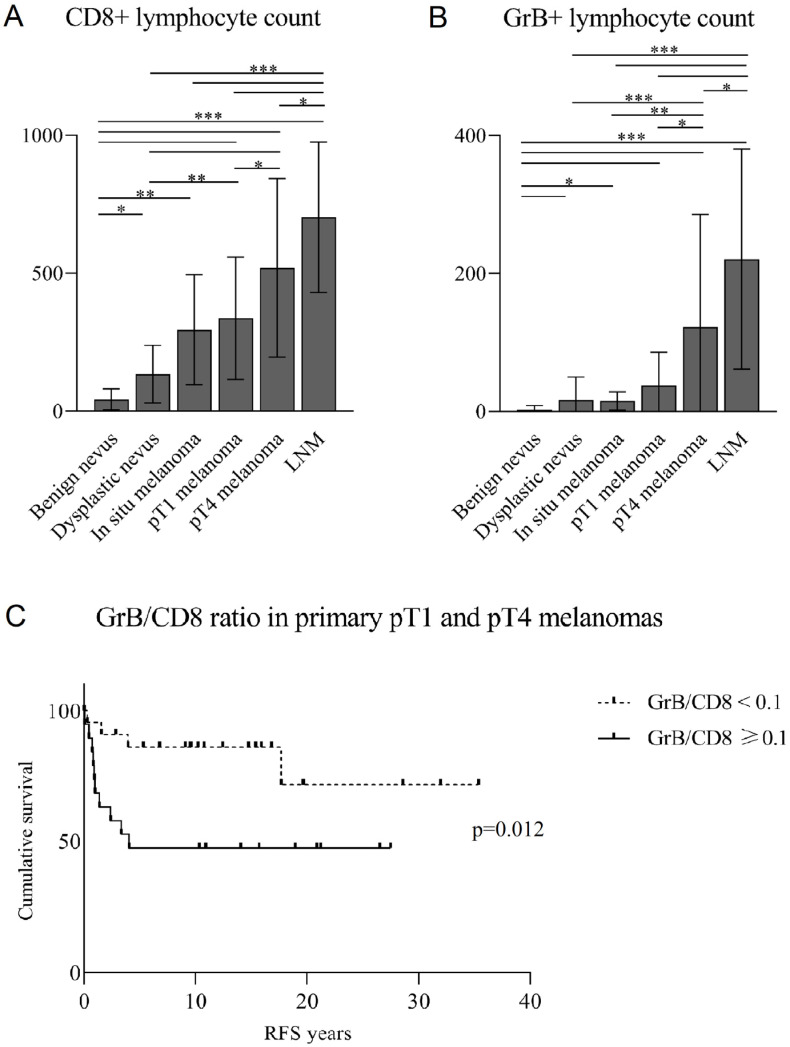
Mean counts of CD8+ lymphocytes (**A**) and GrB+ lymphocytes (**B**) analyzed using hotspot method, and the association of GrB/CD8 lymphocyte ratio with RFS in primary melanomas (**C**). CD8+ lymphocytes were analyzed from 203, and GrB+ lymphocytes from 172 tumor samples. The data in (**A**,**B**) represent mean ± SD. Statistically significant differences between the groups in (**A**,**B**) are shown with connecting lines (Kruskal–Wallis test). * *p* < 0.05, ** *p* < 0.01, *** *p* < 0.001. *p*-value in (**C**) represents the *p*-value obtained by Cox Regression analysis. pT = pathological tumor stage, LNM = lymph node metastasis, RFS = recurrence-free survival.

**Figure 4 biomedicines-10-03209-f004:**
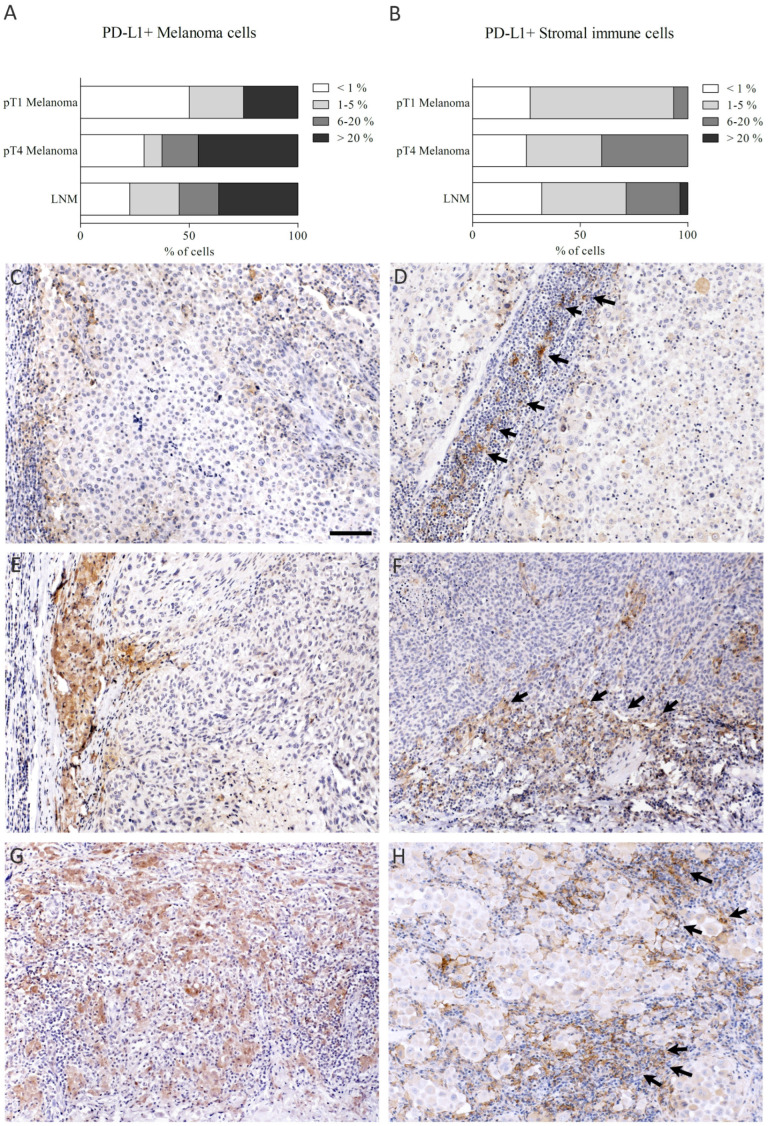
PD-L1 expression in melanoma cells (**A**) and stromal immune cells (**B**) in malignant lesions, and representative immunohistochemical stainings of PD-L1 (**C**–**H**), positive staining shown in brown and counterstained nuclei in blue). PD-L1 expression in melanoma cells and stromal immune cells was evaluated with four-level scoring system. PD-L1 expression 1–5% in tumor and stromal cells in (**C**) and (**D**), 6–20% in (**E**,**F**), and >20% in (**G**,**H**), respectively. Black arrows in (**D**,**F**) and (**H**) mark the PD-L1+ stromal immune cells. Scale bar 100 μm (×100 magnification). pT = pathological tumor stage, LNM = lymph node metastasis.

**Table 1 biomedicines-10-03209-t001:** Sample sizes of analyzed CD8, GrB and PD-L1 stainings. pT = pathological tumor stage, LNM = lymph node metastasis.

	CD8 Stainings *n* (%)	GrB Stainings *n* (%)	PD-L1 Stainings *n* (%)
Benign nevi	27 (13)	27 (16)	
Dysplastic nevi	24 (12)	25 (15)	
In situ melanoma	10 (5)	10 (6)	
pT1 melanoma	36 (18)	29 (17)	14 (21)
pT4 melanoma	46 (23)	37 (22)	28 (41)
LNM	60 (30)	44 (26)	26 (38)
Total	203 (100)	172 (100)	68 (100)

**Table 2 biomedicines-10-03209-t002:** Sample sizes of analyzed CD8 and GrB stainings including clinicopathological data. pT = pathological tumor stage, LNM = lymph node metastasis.

	CD8 Stainings *n* (%)	GrB Stainings *n* (%)
pT1 melanoma	28 (27)	22 (28)
pT4 melanoma	31 (30)	26 (33)
LNM	44 (43)	31 (39)
Total	103 (100)	79 (100)

**Table 3 biomedicines-10-03209-t003:** Clinicopathological characteristics of malignant cases. pT = pathological tumor stage, pN = pathological nodal stage, SD = standard deviation.

Characteristics		*n* (%)
Total number of patients		129
T and N classification		
	pT1	41 (32)
	pT4	41 (32)
	pN1	47 (36)
Age		
	Mean ± SD	59.1 ± 16.6
	Range	5–92
Sex		
	Female	55 (43)
	Male	74 (57)
Breslow’s depth (mm)		
	Mean ± SD	4.5 ± 7.5
	Range	0.2–60.0
Relapse		
	Yes	66 (51)
	No	55 (43)
	Spread at diagnosis	6 (5)
	Missing	2 (2)
Anatomic site of primary melanoma		
	Head and neck	27 (21)
	Trunk	18 (14)
	Back	34 (26)
	Upper limbs	13 (10)
	Lower limbs	22 (17)
	Feet	7 (5)
	Hands	1 (1)
	Fingers or toes	2 (2)
	Not found	4 (3)
Cause of death		
	Malignant melanoma	55 (43)
	Other	17 (13)
	Alive	40 (31)
	Unknown	17 (13)

**Table 4 biomedicines-10-03209-t004:** Spearman’s correlation coefficients for CD8+ and GrB+ lymphocytes and different immunosuppressive factors in malignant melanoma lesions. Tregs = regulatory T cells, TAMs = tumor-associated macrophages.

Variables		IDO+ Stromal Immune Cells	FoxP3+ Tregs	CD68+ TAMs	CD163+ TAMs	GrB+ Lymphocytes
CD8+ lymphocytes	r_s_	0.599	0.357	0.380	0.266	0.829
	*p*-value	<0.001	<0.001	<0.001	0.020	<0.001
GrB+ lymphocytes	r_s_	0.542	0.479	0.429	0.310	
	*p*-value	<0.001	<0.001	<0.001	0.009	

**Table 5 biomedicines-10-03209-t005:** Various associations of CD8+ and GrB+ lymphocytes with clinicopathological parameters.

Variables		CD8+ Lymphocytes			GrB+ Lymphocytes		
		Low *n* (%)	High *n* (%)	*p*-Value	Low *n* (%)	High *n* (%)	*p*-Value
Ulceration				**<0.001**			**0.046**
	Yes	8 (38)	13 (62)		8 (53)	7 (47)	
	No	32 (87)	5 (14)		26 (81)	6 (19)	
Growth pattern				**<0.001**			**0.023**
	Nodular	12 (43)	16 (57)		14 (58)	10 (42)	
	Other	28 (90)	3 (9)		21 (88)	3 (13)	
Presence of mitoses				**0.005**			**0.043**
	Yes	27 (59)	19 (41)		26 (67)	13 (33)	
	No	13 (100)	0 (0)		9 (100)	0 (0)	
Lymph node capsule rupture				**0.003**			**0.035**
	Yes	8 (57)	6 (43)		3 (33)	6 (67)	
	No	4 (14)	25 (86)		1 (5)	20 (95)	
Overall recurrence				**<0.001**			**<0.001**
	Yes	18 (33)	36 (67)		14 (34)	27 (66)	
	No	30 (71)	12 (28)		24 (73)	9 (27)	
Locoregional recurrence				**<0.001**			**<0.001**
	Yes	13 (29)	32 (71)		8 (25)	24 (75)	
	No	37 (66)	19 (34)		31 (69)	14 (31)	
Distal recurrence				**0.021**			0.053
	Yes	19 (38)	31 (62)		15 (40)	23 (61)	
	No	30 (61)	19 (39)		24 (62)	15 (39)	

**Table 6 biomedicines-10-03209-t006:** Associations of PD-L1+ tumor and stromal immune cells with CD8+ and GrB+ lymphocytes and clinicopathological parameters. pT = pathological tumor stage.

Variables		PD-L1+ Tumor Cells			PD-L1+ Stromal Immune Cells		
		<5% *n* (%)	>5% *n* (%)	*p*-Value	<5% *n* (%)	>5% *n* (%)	*p*-Value
CD8+ lymphocytes				**0.015**			**<0.001**
	Low	18 (69)	8 (31)		22 (100)	0 (0)	
	High	15 (39)	24 (62)		20 (54)	17 (46)	
GrB+ lymphocytes				**0.009**			**0.027**
	Low	20 (69)	9 (31)		22 (85)	4 (15)	
	High	10 (35)	19 (66)		16 (57)	12 (43)	
pT classification				**0.037**			**0.034**
	pT1	11 (79)	3 (21)		13 (93)	1 (7)	
	pT4	12 (44)	15 (56)		14 (61)	9 (39)	
Ulceration				0.069			**0.018**
	Yes	6 (38)	10 (63)		7 (50)	7 (50)	
	No	16 (67)	8 (33)		19 (86)	3 (14)	
Growth pattern				**0.009**			**0.001**
	Nodular	10 (40)	15 (60)		11 (52)	10 (48)	
	Other	13 (81)	3 (19)		16 (100)	0 (0)	
Lymph node capsule rupture				**0.032**			0.134
	Yes	7 (70)	3 (30)		8 (80)	2 (20)	
	No	4 (27)	11 (73)		7 (50)	7 (50)	

## Data Availability

The datasets supporting the conclusions of this article are available on request from the authors.

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
