# Peer review of "The Association of CD8+ Cytotoxic T Cells and Granzyme B+ Lymphocytes with Immunosuppressive Factors, Tumor Stage and Prognosis in Cutaneous Melanoma"

_biomedicines, 2022, doi:10.3390/biomedicines10123209_

Round 1

Reviewer 1 Report

The authors intend to demonstrate the association between CD8+, GrB+ and immunosuppressive factors in cutaneous melanoma.

1. The authors should explain in the methodology the differences between the different stages in cutaneous melanoma.

2. In the abstract, they should better focus the conclusion of the work.

3. The introduction should avoid concluding the results of the work.

4. Supplementary tables 1-3 should be incorporated into the manuscript as main tables of the work.

5. Figures and tables must be in line with the text to facilitate reading. The enumeration must be revised.

6. In immunohistochemistry, the different stains and the resulting colors should be explained.

7. Fig3C, the authors should indicate p-log Rank to which time the significance refers.

8. Abbreviations must be included in tables and figures.

Reviewer 2 Report

1.     Please include the IF or IHC  photos of immune suppressive factor staining mentioned in Table 1, besides Foxp3 in Fig 2, if the data were generated in this manuscript; otherwise, please cite the references.

2.     NK and NKT  cells are another major source of GrB in TME, I wonder if the authors checked the NK/NKT cell and GrB ratio in the research?

3.     Could the authors find the cytotoxic CD4  T cells in the TME, which generates GrB?

4.     Are the GrB counts associated with the dysfunctional CD8 TILs in this cohort?

5.     Fig 4 is disconnected from the previous figs. 

Reviewer 3 Report

biomedicines-2031315-peer-review-v1

The association of CD8+ cytotoxic T cells and Granzyme B+ lymphocytes with immunosuppressive factors, tumor stage and prognosis in cutaneous melanoma 

Granzyme B (GrB) conveys cytolytic activities of CD8+ cytotoxic lymphocytes but is also expressed by immunosuppressive cells for immunosuppressive purposes.  You could mention the type of immunosuppressive cells

Benign, premalignant, and malignant melanocytic tumors were stained immunohistochemically for CD8 and GrB. PD-L1 was also stained from malignant samples containing clinicopathological data. What was the reason all 3 types of samples weren’t also stained for PD-L1?

Our aim was to obtain a more comprehensive perception of the immunosuppressive TME and study the role of GrB+ lymphocytes in melanoma.  The second section of this sentence is not clear, was the role of GrB+ also linked to TME immunosuppression?

Surprisingly, a low GrB/CD8 ratio was associated with better recurrence-free survival in primary melanomas. In the following sentence “CD8+ and GrB+ lymphocytes associated positively with PD-L1+ and IDO+ tumor and stromal immune cells”. It is interesting that low GrB cells are linked to better survival yet are associated with immunosuppression of PDL1 and IDO, is this slightly contradictory?

Our results suggest that GrB+ lymphocytes might represent more activated immunosuppressive than cytotoxic lymphocytes.  Would be interesting to see how you show that there are indeed two categories here.

CD8+ CTLs also associated with favorable prognosis in several studies [8], in both primary [9] and advanced melanomas [10-11], and may also predict response to ICIs [12] However, in some studies, CTLs have not been shown to associate 48 with CM prognosis. What is the reason for the discrepancy?

Granzyme B (GrB) is a serine protease excreted through exocytosis by CTLs to kill 50 tumor cells [14]. Thus, GrB has been used as a marker for anti-tumor cytolytic activity for 51 examples in colorectal cancer, where it associates positively with survival [15-16]. How- 52 ever, GrB is also expressed by immunosuppressive cells, like FoxP3+ Tregs and B regula- 53 tory cells (Bregs) to mediate their immunosuppressive effects. Please explain why the same serine protease can supposedly have contrastingly different roles depending on the cell producing it.

The prognostic significance of PD-L1 in CM remains unclear. In some studies, PD- 61 L1+ tumor or stromal cells associated with poor outcomes [20-21], while others have not 62 found any association between PD-L1 expression and prognosis [11,22]. What is the reason for this contradiction Please explain in the text.

The purpose of this study was to investigate the associations of CD8+ and GrB+ lym- 64 phocytes. Please add a few notes about the difference between these two types.

Our results suggest that the amount of tumor-infiltrating CD8+ and GrB+ lympho- 69 cytes increase with the tumor malignancy. IDO and PD-L1 seem to be key immunosup- 70 pressive factors in CD8+ lymphocyte predominant tumors. However, GrB+ lymphocytes 71 seem to represent cytolytic activity of immunosuppressive lymphocytes rather than CTLs, 72 and their amount seem to associate with the accumulation of several other immunosup- 73 pressive cells into the tumor. The point above again that please define the CD8+, GrB+ and CTL groups to make sure all your readers are on the same page since these terms (and their differences) are key to understanding this study.

From a total 78 of 252 samples, 250 were stained for CD8 and 249 for GrB. PD-L1 was stained from 128 79 malignant tumors containing patient data. Why weren’t they all stained with all three antibodies?

For the statistical tests, please inform the readers if any correction for multiple testing including FDR/ Bonferroni correction was performed. 

In section 3.1. please give a brief summary of the general characteristics of these patients; what stage of CM, before or after treatment, gender and age distribution. I realise that this has been done in Supplementary file 3 but please give some insight in writing as well.

To evaluate the density and localization of CD8+ and GrB+ lymphocytes, melanocytic 174 tumors were stained for CD8 and GrB. The staining pattern of CD8 was cytoplasmic and 175 membranous. GrB showed granular, cytoplasmic staining pattern and was also localized 176 to the plasma membrane similarly as CD8 (Figure 1 and 2). Both CD8+ and GrB+ lympho- 177 cytes localized mainly in the stromal compartment. It would be good for the authors to distinguish a protein that is secreted such as granzymes compared to an integral membrane protein. It would make sense that GrB is granular and cytoplasmic and CD8 is both plasma membrane-bound and cytoplasmic. Since all these proteins are assembled in the endoplasmic reticulum, so at any point in time, there will be a stream of these proteins making their way from the ER to the membrane or ECM enclosed in vesicles.

Please give a narrative for figures 1 and 2 in the main text (i.e., levels of CD8 and GrB+ cells depending on CM classification).

The number of CD8+ and GrB+ lymphocytes was significantly higher in pT4 com- 181 pared to pT1 melanomas (p-values 0.023 and 0.014, respectively) (Figure 3 A, B).  How about samples that are metastatic? How do these CD8+ and GrB+ lymphocytes fair?

CD8+ and GrB+ lymphocytes were associated positively with recurrence (p-values <0.001) 186 and poor prognostic factors, like ulceration (p-values <0.001 and 0.046, respectively) and 187 nodular growth phase (p-values <0.001 and 0.023, respectively) (Table 2). Please narrate this finding to distinguish the similarities and differences between the CD8+ and GrB+ lymphocytes links with these clinical attributes.

In the group of all malignant lesions, high CD8+ and GrB+ lymphocyte counts asso- 195 ciated with poor recurrence-free survival (RFS) in univariate survival analysis (p-values 196 <0.001 and 0.002, respectively), but the result was not retained in multivariate analysis 197 with tumor stage as a covariate. In the group of primary melanomas only, high CD8 count 198 associated with poor RFS (p=0.015) and poor disease-specific survival (DSS) (p=0.037) but 199 was not an independent prognostic factor when Breslow’s depth was used as a covariate. In the group of all malignant cases and primary melanomas only, GrB/CD8 ratio <0.1 201 was associated with better RFS (p-values <0.001 and 0.019, respectively). How about the presence of GrB+ cells only? 

Tumors were divided into low (≤5 % of cells) and high (>5 % of cells) PD-L1 express- 210 ing tumors. PD-L1+ tumor and stromal immune cells were significantly more abundant 211 in pT4 compared to pT1 tumors (p-values 0.037 and 0.034, respectively) (Table 3). No sta- 212 tistically significant differences between PD-L1 expression in different tumor stages was 213 observed when PD-L1 was evaluated in four-level scoring system (Figure 4 A, B).  How to the authors explain the association of PD-L1 with TNMs but not stage?

The correlations of CD8+ and GrB+ lymphocytes with immunosuppressive factors 222 (PD-L1, IDO, FoxP3 Tregs and TAMs), were evaluated from malignant tumors. CD8+ 223 CTLs associated positively with PD-L1+ tumor and stromal immune cells (p-values 0.015 224 and <0.001, respectively) (Table 3), IDO+ melanoma cells (p<0.001) and tumor nest CD68+ 225 macrophages (p=0.016) (data not shown). There was a moderate positive correlation be- 226 tween CD8+ CTLs and IDO+ stromal immune cells (rs=0.599, p<0.001), but only weak pos- 227 itive correlations with FoxP3+ Tregs and total macrophage counts (Table 1). 228 

GrB+ lymphocytes associated positively with PD-L1+ tumor and stromal immune 229 cells (p-values 0.009 and 0.027, respectively) (Table 3), IDO+ melanoma cells (p<0.001), and 230 tumor nest CD68+ and CD163+ macrophages (p-values 0.002 and 0.003, respectively) (data 231 not shown). There was a moderate positive correlation of GrB+ lymphocytes with IDO+ 232 stromal cells (rs=0.542, p<0.001) and FoxP3 Tregs (rs=0.479, p<0.001) (Table 1). In immuno- 233 fluorescence double stainings, GrB+ cells localized in the same region as FoxP3+ Tregs in 234 the tumor stroma interface but no clear colocalization was observed (Figure 2 G). 

Please narrate the difference and similarities between GrB+ and CD8+ cells with immunosuppressive cells as a summarising sentence at the end of this section (3.6).  For example, is the link stronger in GrB+s?

PD-L1 expression did not associate with 240 FoxP3+ Tregs or TAMs. This is interesting that various immunosuppression facets are not necessarily linked.

In general, the narrative of the results section needs expansion. 

Figure 3 is very useful since its shows a visible/ quantified representation of the data that is easily understood. Figure 3A-B is interesting that both cell types show a similar trend.

In figure 3C, please also display any links to survival for CD8 and GrB+ cells individually.

Please narrate the significance of figure 4a-b.

Surprisingly, low GrB/CD8 ratio (<0.1) associated with better non-immunotherapy 275 related RFS in primary melanomas. In colorectal cancer, GrB+ cells have been shown to 276 associate with better prognosis and this has been used as a cytolytic marker for anti-tumor 277 immunity [15-16]. Any precedence in melanoma specifically?

Moreover, the present work demon- 272 strates that CD8+ and GrB+ lymphocytes associate positively with PD-L1+ and IDO+ tu- 273 mor and stromal immune cells in the TME.  Specify the type of stromal immune cells in this sentence.

The present result indicating that low amount of GrB+ lymphocytes 278 with respect to tumor infiltrating CTLs associates with better RFS may refer to the immu- 279 nosuppressive role of GrB in CM. Indeed, GrB is also expressed by a variety of immuno- 280 suppressive cells, like Bregs and Tregs [17]. For example, Sabbatino et al. analyzed im- 281 mune cells in thin melanomas and found that GrB+ cells did not colocalize with CD8+ cells 282 in double stainings [26]. Please tell the readers, how this is mechanistically possible for GrB expressed by other cells (B regs and T regs) to have a completely opposite effect. How is this justifiable enzymatically?

Wu et al. used a single-sample gene set enrichment analysis to assess the role of 298 granzymes in CM and found that GrB is associated positively with immunotherapy-related 299 prognosis [27].  Which is why I asked to add this info to figure 4C.

We also found that GrB+ lymphocytes associate positively with IDO+ melanoma cells 318 and tumor nest macrophages and moderately with FoxP3 Tregs and IDO+ stromal im- 319 mune cells. Similar associations have not been studied before in CM. Our results suggest 320 that GrB+ lymphocytes might represent mainly immunosuppressive lymphocytes and 321 thus, the positive association of GrB+ lymphocytes with different immunosuppressive fac- 322 tors may indicate that the activation of immunosuppressive lymphocytes is associated 323 with a concurrent accumulation of other immunosuppressive factors into the tumor. 324 However, further studies are needed to assess the role of GrB+ lymphocytes in CM.  Please explain the potential mechanism.

CD8+ and GrB+ lym- 20 phocytes were more abundant in pT4 compared to pT1 melanomas, and in lymph node metastases 21 compared with primary melanomas. Surprisingly, low GrB/CD8 ratio was associated with better 22 recurrence-free survival in primary melanomas. furthermore, CD8+ and GrB+ lymphocytes associ- 23 ated positively with PD-L1+ and IDO+ tumor and stromal immune cells and tumor nest CD68+ 24 macrophages. This is me looking at your abstract again, how can Grb+ cell increase with pT levels yet low GrB/CD8 ratios are linked with better survival? In one case GrB+ are immunoreactive (pT) and in the next case, they are immunosuppressive (when talking about the ratio)? Or in both scenarios they are immunosuppressive but the difference is in the presence of CD8+ cells for example that are immunoreactive and eliminates the tumour cells. If so, why do the CD8+ cells show the same trend between pT1-T4? Yet both factors are linked to immunosuppression most notably GrB+? Please explain this in your discussion more thoroughly. Also, I would ask the authors to draw a cartoon of their working hypothesis to explain this (I assume the working hypothesis is this: Our results suggest that GrB+ lymphocytes might represent more activated immunosuppressive than cytotoxic lymphocytes). there doesn't seem to be a lot of supporting literature on CM, so the visual representation of the results of this study will really help lay the groundwork for future studies.

Round 2

Reviewer 1 Report

The article is correct in the present form

Reviewer 2 Report

The authors have addressed my questions.

Reviewer 3 Report

The authors have addressed my comments, thanks.